# Worldwide genetic variation of the IGHV and TRBV immune receptor gene families in humans

Shishi Luo[1,2,*], Jane A Yu[1,*], Heng Li[3] , Yun S Song[1,2,4]

**The immunoglobulin heavy variable (IGHV) and T cell beta variable (TRBV) loci are among the most complex and variable regions in the human genome. Generated through a process of gene duplication/deletion and diversification, these loci can vary extensively between individuals in copy number and contain genes that are highly similar, making their analysis technically challenging. Here, we present a comprehensive study of the functional gene segments in the IGHV and TRBV loci, quantifying their copy number and single-nucleotide variation in a globally diverse sample of 109 (IGHV) and 286 (TRBV) humans from over a 100 populations. We find that the IGHV and TRBV gene families exhibit starkly different patterns of variation. In addition to providing insight into the different evolutionary paths of the IGHV and TRBV loci, our results are also important to the adaptive immune repertoire sequencing community, where the lack of frequencies of common alleles and copy number variants is hampering existing analytical pipelines.**

## Introduction

By some estimates, genomic variation due to copy number differences underlies more variation in the human genome than that due to single-nucleotide differences (Tuzun et al, 2005; Sudmant et al, 2015). Yet, copy number variation remains challenging to quantify and analyze. Nowhere is this more true than in genomic regions that contain gene families: collections of genes formed through the process of duplication/deletion and diversification of contiguous stretches of DNA (Nei & Rooney, 2005). Two gene families that are of particular biomedical relevance but for which variation is not well characterized are the immunoglobulin heavy variable (IGHV) family, a 1-Mb locus located on chromosome 14 (Matsuda et al, 1998; Watson et al, 2013), and the T-cell receptor beta variable (TRBV) family, a 500-kb locus located on chromosome 7 (Rowen et al, 1996). Both regions undergo VDJ recombination, providing the V (variable) component in the biosynthesis of adaptive immune receptors: the IGHV for the heavy chain of the B-cell receptor and the TRBV for the beta chain of the T-cell receptor (Murphy & Weaver, 2016). In the human genome, both loci are organized as a series of approximately 45 functional V gene segments and are adjacent to a collection of D (diversity) and J (joining) segments. Both loci are present in the genomes of all vertebrates known to have an adaptive immune system, although the arrangement of the IGHV locus can differ between species (Cannon et al, 2004; Das et al, 2008; Flajnik & Kasahara, 2010). Indeed, the genes comprising the IGHV and TRBV loci are distant paralogs and are believed to derive from a common ancestral locus in a vertebrate contemporaneous with or predating jawed fishes (Cannon et al, 2004; Das et al, 2008; Flajnik & Kasahara, 2010). That these two loci share genomic features and evolutionary origins makes them an ideal system for a comparative study in gene family evolution.

Here, we present the largest investigation to date of genetic variation in the IGHV and TRBV loci using short-read whole-genome sequencing data. We apply a customized genotyping pipeline (based on Luo et al [2016]) to data from the Simons Genome Diversity Project (SGDP) Mallick et al (2016), which performed whole-genome sequencing of a globally diverse sample of human individuals from over a 100 populations. Such characterization of population-level genetic variation in the immune receptor loci sheds light on how the two loci evolved from their common origins. Quantification of variation is also needed in the burgeoning field of computational immunology (Robins, 2013; Georgiou et al, 2014), where the relative abundances of germline variants will help in other applications such as genome-wide association studies, measuring linkage disequilibrium, and determining clonal lineages from VDJ sequences. For example, previous work demonstrates that the V genes may contribute a significant proportion of the CDR3, and oftentimes, lineages with conserved D and J genes must be distinguished using V gene information (Li et al, 2004). Past methods for CDR3 determination have included integrating over all possible V genes when information was lacking and taking population-wide frequencies into account would likely improve the accuracy of such methods (Murugan et al, 2012). In addition, the common copy number polymorphisms we find in our data agree with what has previously been documented, and the most frequent allele we

[1]Computer Science Division, University of California, Berkeley, Berkeley, CA, USA   [2]Department of Statistics, University of California, Berkeley, Berkeley, CA, USA   [3]Department of Biostatistics, Harvard Medical School, Boston, MA, USA   [4]Chan Zuckerberg Biohub, San Francisco, CA, USA

Correspondence: yss@berkeley.edu
*Shishi Luo and Jane A Yu contributed equally to this work.

report for each gene segment corresponds to the first or second allele (*01 and *02, respectively) recorded for that gene segment in the ImMunoGeneTics information system (IMGT) database (Giudicelli et al, 2005). We emphasize, however, that the results from this genotyping is used purely for aggregate measures of sample-level variation. Our method is not intended to be used to accurately genotype individual genomes.

# Results

A brief note about gene nomenclature: for the bioinformatic analysis, it was necessary to group together gene segments that are operationally indistinguishable but which have distinct names because they occupy physically different positions in the genome. Our departure from this standard nomenclature is detailed in the Materials and Methods section and is also explained where needed below.

To minimize confusion around terminology, we use *polymorphism* as a general term for a genomic unit (nucleotide position or gene segment) that exhibits variation between genomes. Different instances of a particular polymorphism are called *variants*, for example, a single-nucleotide variant (SNV) or a gene copy number variant (CNV). In line with the usage in the immunogenetics community, the term *allele* is reserved exclusively for referring to variants of a gene, as in the allele *IGHV1-69*01*, which is a gene-length variant of the *IGHV1-69* gene segment and which may differ in more than a single nucleotide from other alleles of *IGHV1-69*. We use *haplotype* to refer to the set of operationally distinguishable gene segments that are inherited from a single parent.

**Copy number variation**

In general, gene duplication/deletion appears to have occurred more frequently in the IGHV locus than in the TRBV locus. This is evident in the greater variation in the number of operationally distinguishable IGHV gene segments than in TRBV gene segments (Figs 1 and S1). Using our per-segment copy number estimates and hierarchical clustering (see Supplementary Text, Figs S2–S6), we identified locus-wide copy number haplotypes, some of which have been previously reported (Figs 2 and 3). This and the more detailed figures in Supplementary Information will serve as a useful reference for the computational immunology community. To be conservative, we restricted our figure results to polymorphisms that either involve at least two operationally distinguishable gene segments or involve a single gene segment with high levels of copy number variation. Several IGHV genes (*IGHV7-4-1*, *IGHV4-4*, *IGHV4-30-4*, *IGHV4-59*, and *IGHV4-61*) had unusual read-coverage profiles and, to be conservative, were not included in the CNV calls. In contrast, TRBV genes had predominantly well-behaved read coverages and were two-copy per individual, resulting in a more complete list of CNVs in TRBV (Fig 3).

**Lack of geographical associations**
We considered grouping individuals according to the geographic regions defined by SDGP, namely, Africans, West Eurasians, Central

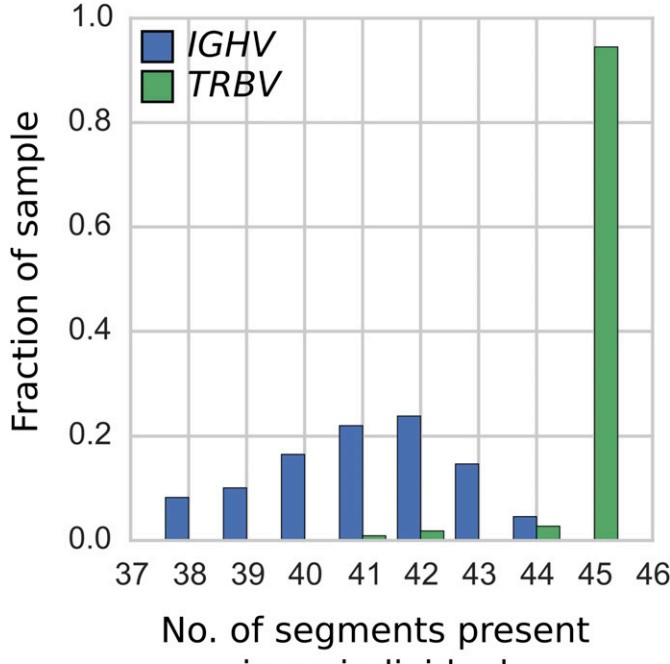

**Figure 1. Histogram of the number of gene segments in an individual.**
The results are based on the IGHV (blue) and TRBV (green) segments present in each of the 109 individuals from blood and saliva samples. The number of operationally distinguishable IGHV gene segments shows greater variation than the number of TRBV gene segments. Fig S1 shows a histogram of the number of TRBV gene segments in the full set of 286 individuals.

Asians–Siberians, East Asians, South Asians, Oceanians, and Native Americans. In most cases, we found that the distribution of CNVs within a geographic region is consistent with the global distribution (Figs S7 and S8). The two exceptions are (i) the polymorphism involving *IGHV1-69*, where the duplication/insertion variant is the major variant among genomes sampled from Africa, despite being a minor variant (28%) of the global sample, and (ii) the three-gene deletion of *TRBV5-8*, *TRBV7-8*, and *TRBV6-9*, which is the major variant among genomes sampled from the Americas but appears in only 5% of our sample globally. In neither of these two cases is there evidence to suggest the absence of any particular gene is fatal. We note, however, that the sample sizes for the IGHV analysis of East Asians, Oceanians, and Native Americans do not have suitable statistical power and are included for comprehensiveness and illustrative purposes.

*No correlation between copy number polymorphisms*
We found effectively no correlation between copy number polymorphisms in either IGHV or TRBV (Figs S9 and S10). The average value of $R^2$, the square of the Pearson correlation coefficient, between segments in the different polymorphisms is 0.021 for the IGHV gene segments (Fig 2) and 0.004 for the TRBV gene segments (Fig 3). Thus, the polymorphisms are essentially independent, and we can estimate the number of copy number haplotypes in the two loci. From Fig 2, with three polymorphisms each with two (haploid) variants, and with the set {*IGHV3-30*, *IGHV3-30-3*, *IGHV3-30-5*, *IGHV3-33*} and {*IGHV3-23*, *IGHV3-23D*} exhibiting an estimated seven and

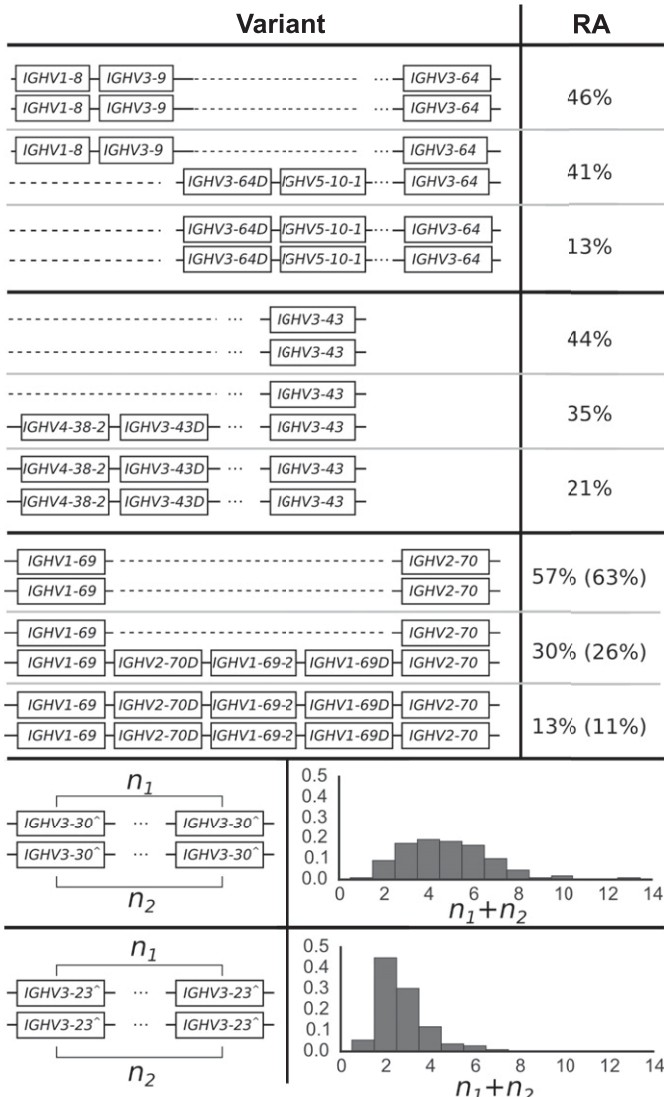

| Variant | RA |
|---|---|
| IGHV1-8 – IGHV3-9 ········· ··· IGHV3-64 / IGHV1-8 – IGHV3-9 ········· ··· IGHV3-64 | 46% |
| IGHV1-8 – IGHV3-9 ········· IGHV3-64 / ········· IGHV3-64D–IGHV5-10-1···· IGHV3-64 | 41% |
| ········· IGHV3-64D–IGHV5-10-1···· IGHV3-64 / ········· IGHV3-64D–IGHV5-10-1···· IGHV3-64 | 13% |
| ········· ··· IGHV3-43 / ········· ··· IGHV3-43 | 44% |
| ········· ··· IGHV3-43 / IGHV4-38-2–IGHV3-43D ··· IGHV3-43 | 35% |
| IGHV4-38-2–IGHV3-43D ··· IGHV3-43 / IGHV4-38-2–IGHV3-43D ··· IGHV3-43 | 21% |
| IGHV1-69 ········· IGHV2-70 / IGHV1-69 ········· IGHV2-70 | 57% (63%) |
| IGHV1-69 ········· IGHV2-70 / IGHV1-69–IGHV2-70D–IGHV1-69-2–IGHV1-69D–IGHV2-70 | 30% (26%) |
| IGHV1-69–IGHV2-70D–IGHV1-69-2–IGHV1-69D–IGHV2-70 / IGHV1-69–IGHV2-70D–IGHV1-69-2–IGHV1-69D–IGHV2-70 | 13% (11%) |

**Figure 2.   The distribution of IGHV copy number polymorphisms reliably called in our sample.**
Schematics in the left column show the polymorphisms, whereas the right column displays the relative abundance (RA) in the sample of 109 individuals. Our data inform the copy number of these genes, whereas the genomic configuration is our best estimate based on previous studies (Sasso et al, 1993; Milner et al, 1995; Sasso et al, 1996; Cho et al, 1997; Pramanik & Li, 2002; Chimge et al, 2005; Boyd et al, 2010; Watson et al, 2013). For the polymorphism involving IGHV1-69, we also show the relative abundances in the full sample of 286 individuals in parentheses. This is because IGHV1-69 and IGHV2-70 are located in the J-distal part of the IGHV locus, making them less likely to be affected by VDJ recombination. Unlike the IGHV polymorphisms that are closer to the J region, we saw negligible differences in copy number estimates for these gene segments in the saliva versus cell-line samples. Note that we use IGHV3-30ˆ as shorthand for the set {IGHV3-30, IGHV3-30-3, IGHV3-30-5, IGHV3-33} and IGHV3-23ˆ for {IGHV3-23, IGHV3-23D}. The relative abundances for the CNVs of IGHV3-30ˆ and IGHV3-23ˆ are for the total number summed over the two haplotypes in an individual. We estimate the largest haploid number to be 7 for IGHV3-30ˆ (because the smallest two numbers to sum to 13, the largest copy number called, are 6 and 7) and 4 for IGHV3-23ˆ (likewise, 4 and 3 are the smallest two numbers to sum to 7). IGHV7-4-1 had inexplicable inflated coverage that prevented us from applying the same analysis to determine its population frequencies. See Supplementary Information.

| Variant | RA |
|---|---|
| TRBV4-2 – TRBV6-2 ············ / TRBV4-2 – TRBV6-2 ············ | 42% |
| TRBV4-2 – TRBV6-2 ············ / TRBV4-2 – TRBV6-2 – TRBV4-3 – TRBV6-3 | 43% |
| TRBV4-2 – TRBV6-2 – TRBV4-3 – TRBV6-3 / TRBV4-2 – TRBV6-2 – TRBV4-3 – TRBV6-3 | 15% |
| TRBV5-8 – TRBV7-8 – TRBV6-9 / TRBV5-8 – TRBV7-8 – TRBV6-9 | 91% |
| TRBV5-8 – TRBV7-8 – TRBV6-9 / ············ | 8% |
| ············ / ············ | 1% |

**Figure 3.   The distribution of TRBV copy number polymorphisms reliably called in our sample.**
Schematics in the left column show the polymorphisms, whereas the right column displays the relative abundance (RA) in the full sample of 286 individuals. Our data inform the copy number of these genes, whereas the genomic configuration is our best estimate based on previous studies. The insertion of {TRBV4-2, TRBV4-3} and {TRBV6-2, TRBV6-3} is a frequent polymorphism also found in previous studies (Seboun et al, 1993; Zhao et al, 1994; Subrahmanyan et al, 2001). The polymorphism involving TRBV5-8, TRBV7-8, and TRBV6-9 was identified by first clustering using TRBV5-8 copy number estimates alone, and then noticing that such a clustering also induced a clear-cut partition of the copy number estimates for TRBV7-8 and TRBV6-9. See Supplementary Information.

four (haploid) CNVs, respectively, this gives approximately 200 IGHV haplotypes (2 × 2 × 2 × 7 × 4), assuming independence between the common copy number polymorphisms. The analogous calculation from Fig 3 for TRBV leads to only a handful of haplotypes (2 × 2). We note that this result is not meant to be taken literally. Rather, the orders of magnitude difference between our estimates for IGHV haplotypes compared with TRBV haplotypes strongly suggests that the two loci have undergone different rates of gene duplication and deletion.

## SNV and allelic variation in two-copy gene segments

Having quantified copy number variation of gene segments across the two loci, we sought to compare nucleotide variation while minimizing the confounding factor of copy number variation. A gene segment with higher copy number could be perceived as exhibiting greater single-nucleotide or allelic variation, even though it experiences the same rate of per-base substitution. For this reason, we compared single-nucleotide and allelic variation in IGHV and TRBV gene segments that have two copies in the vast majority

of individuals in our sample and for which there is minimal read-mapping ambiguity (11 such IGHV segments, 40 TRBV segments, see Supplementary Text). We will refer to such gene segments as "two-copy" for short. In this context, single-nucleotide polymorphisms (SNPs) are meant to refer to nucleotide positions that are polymorphic when compared across individuals in our sample, whereas a SNV is a specific genetic type occurring at an SNP. This is in contrast to a "novel allele," which refers to a sequence of nucleotides that does not exactly match any known allele in the IMGT database. In addition to restricting our single-nucleotide and allelic analysis to two-copy gene segments, we were also conservative in how we called these variants: an allele or SNV is called only if it is present in two or more individuals. To be clear, the only analysis that is limited to 11 IGHV genes (as opposed to 40 TRBV genes) is the allelic/SNP variation analysis.

### IGHV and TRBV have comparable levels of nucleotide diversity in two-copy genes

We find that when restricted to the set of two-copy gene segments in IGHV and TRBV, the two loci have comparable summary measures of single-nucleotide and allelic variation (Table 1, Fig 4). If anything, the TRBV two-copy gene segments exhibit greater single-nucleotide and allelic diversity, given the higher number of SNVs and average base pair differences between the 109 individuals. We find that on average, IGHV two-copy gene segments have 1.7 SNVs, as opposed to 1.9 SNVs per TRBV two-copy gene, which is similar to previous work reporting roughly two SNVs per gene (Mackelprang et al, 2002; Subrahmanyan et al, 2001). Our slight underestimate of this value is reasonable given that we restrict our analysis to two-copy genes. That TRBV exhibits greater or comparable diversity is seemingly surprising because if allelic diversity is estimated by taking the average number of alleles per gene segment as per the IMGT database, without regard to the segment's copy number, operationally distinguishable IGHV gene segments have an average of five alleles, whereas TRBV gene segments have an average of two alleles. This discrepancy in the two ways of estimating sequence variation does not seem to be due to any under-representation of TRBV alleles in the IMGT database relative to IGHV alleles: the fraction of putative novel alleles called in our sample is similar between the IGHV and TRBV gene segments (Table 1, third row). However, if high copy-number segments were included in this allelic diversity analysis, then the per-segment allelic diversity for the IGHV locus would likely be higher than the observed diversity for two-copy segments, as suggested by the results of the work by Scheepers et al (2015). This discrepancy could indicate that our observation of elevated nucleotide diversity in two-copy gene segments may not hold for the

IGHV and TRBV loci as a whole; it could be that restricting to two-copy gene segments filters out IGHV genes with higher levels of nucleotide diversity, which could have resulted from relaxed selective pressure in higher copy-number gene segments. As a reference for the antibody repertoire sequencing community, we have provided the relative abundances of alleles for the two-copy gene segments calculated from our sample in Tables S1 and S2.

### Putatively novel alleles

We called 28 IGHV alleles, of which 12 are putatively novel, and 97 TRBV alleles, of which 38 are putatively novel. Of these novel alleles, it is notable that 5 IGHV alleles and 12 TRBV alleles appeared at least 10 times in our sample (we count homozygous alleles as appearing twice; Table S3). Some of these novel alleles such as *IGHV1-45*02_ga123 GR*, *TRBV10-1*02_gt234E_*, and *TRBV12-5*01_cg27HD* (see Supplementary Text for information on this notation) are present in high frequency across all geographic regions. That these novel variants are comprehensively present supports existing evidence that the databases of IGHV and TRBV alleles are not yet complete (Gadala-Maria et al, 2015; Scheepers et al, 2015; Corcoran et al, 2016; Gidoni et al, 2019).

### SNV and allelic variants private to geographic regions

An SNV that is private to a geographic region indicates that individual(s) all from one region have a base pair that differs from the base pair of all other individuals at that site. Alternatively, an allele that is private to a geographic region indicates that an entire allelic sequence is specific to individual(s) from that region. We found 5 SNVs in the 11 two-copy IGHV gene segments that are private to a single geographic region and 14 such variants in the 40 two-copy TRBV gene segments (Table S4). These variants are not rare: most of them are present at greater than 10% frequency in the geographic region that they are private to, with the extremes being as high as 42%. For both loci, the geographic region of Africa had a disproportionate share of such variants: of the five IGHV SNVs that were private to a geographic region, all five were private to Africa and of the 14 SNVs exclusive to a region for TRBV, 10 (71.4%) were private to Africa. This particular feature of samples from the Africa region is also apparent in our allelic variation analysis (Table S5). Of the 28 IGHV alleles we called, 4 out of 4 private alleles were private to Africa. Similarly, of the 97 TRBV alleles we called, 10 out of 14 (71.4%) private alleles were private to Africa. These findings of higher levels of diversity primarily in Africa are consistent with prior studies (Mackelprang et al, 2002; Conrad et al, 2006; Jakobsson et al, 2008; Li et al, 2008; Ramachandran et al, 2005; Tishkoff & Kidd, 2004; Tishkoff & Verrelli, 2003; Tishkoff & Williams, 2002; Zhao et al, 2006) and with

**Table 1. Summary statistics for single nucleotide and allelic variation in IGHV and TRBV.**

|  | IGHV | TRBV |
|---|---|---|
| Average bp difference per pairs of alleles | 4.1% | 5.0% |
| Average no. of SNPs per segment | 1.7 | 1.9 |
| Fraction of novel alleles out of all observed alleles | 12/28 (43%) | 40/99 (40%) |

The results tabulated are computed using the same set of 109 individuals and are restricted to the two-copy segments described in the text. To calculate the average base pair difference per pairs of alleles, for each segment, we computed the average base pair difference between all pairs of alleles, and then averaged over all segments.

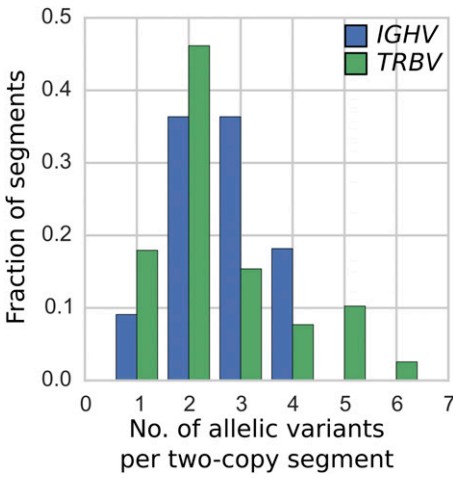

**Figure 4. The number of alleles in the 11 two-copy IGHV (blue) and 40 two-copy TRBV (green) segments.**
We report an allele only if it is found in at least two out of the 109 genomes from blood and saliva samples. The two distributions are not statistically significantly different (*P* value of two-sample Kolmogorov–Smirnov test between the blue and green distribution is 0.97).

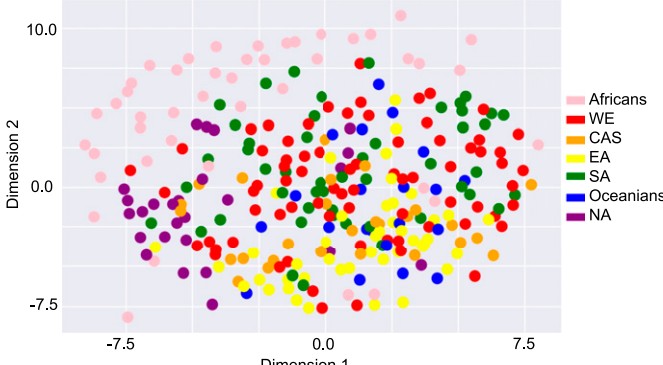

**Figure 5. Multidimensional scaling of TRBV alleles.**
Based on 286 individuals from all populations (including all DNA source types) in the SGDP dataset. Each point corresponds to an individual and is colored by the corresponding geographic region defined by SDGP: Africans, West Eurasians (WE), Central Asians–Siberians (CAS), East Asians (EA), South Asians (SA), Oceanians, and Native Americans (NA). Multidimensional scaling was performed in Python using the manifold.MDS([n_components, metric, n_init,s..]) function from the sklearn. manifold module. The data fit by the model uses the Euclidean distance between $x_i$ and $x_j$ where the $m$th entry in vector $x_i$ is the copy number of allele m in individual $i$, taking possible values 0, 1, or 2.

the percentage of exon-located SNVs that are private to Africa across the entire genome (72.1%). For a complete table of SNVs and alleles private to a particular region, see Tables S3 and S4.

### Geographical clustering of TRBV haplotypes

To investigate whether genetic variation at immune receptor loci exhibits geographical structure, we applied multidimensional scaling to the reconstructed phased TRBV haplotypes from 286 individuals for each two-copy gene. Fig 5 illustrates our result, where each point corresponds to an individual. Supplementary Information includes results from applying multidimensional scaling to individuals just from pairs of geographic regions.

As shown in Fig 5, we observed the clearest separation between the African population and the rest of the populations, a trend that is also apparent in the pairwise plots (Supplementary Information) and is related to our aforementioned finding that African individuals tend to have the most alleles private to one region. There are a few African individuals who are exceptions to this pattern. Specifically, Masai-1 from Kenya and Saharawi-2 from Morocco consistently cluster more closely with Eurasians.

Although distinction amongst the other populations is not immediately obvious from Fig 5, every pairwise comparison with Native Americans showed reasonably clear separation from the other populations, which may be due to the reduced genetic diversity of Native Americans compared with that of other populations (Wang et al, 2007). In addition, the individuals from Central Asia—Siberia and South Asia were fairly separable, although the degree of distinction is less prominent compared with those discussed above. Comparisons demonstrating significant overlap include Central Asia—Siberia versus East Asia, West Eurasia versus Central Asia—Siberia, and West Eurasia versus South Asia, which is expected given previous reports of high gene flow between Europe and Asia (Qin et al, 2015). Given these results, we would expect high fixation index values between each subpopulation and Africa/

Native America and lower fixation index values otherwise, which is indeed what we found (Table S5).

### General variation patterns suggest distinct evolutionary dynamics

Our analysis of all functional operationally distinguishable gene segments in the two loci indicates more gene duplication/deletions in IGHV than in TRBV (Fig 1). In contrast, the observed level of nucleotide diversity within gene segments—as measured by the amount of sequence variation per gene segment in two-copy genes—seems to be slightly higher in the TRBV locus than in the IGHV locus (Table 1, Fig 4). If the rate of sequence diversification were indeed higher in TRBV than in IGHV, we would expect the IGHV gene family to comprise genes that are more similar to each other on average than the TRBV gene family. This holds true for the genes found in the IMGT annotated gene table for humans. For all pairs of functional genes, we measured between-segment diversity as the pairwise global alignment score (see Supplementary Text for details) between gene segments, which gives significantly higher scores for the IGHV genes, indicating more mismatches and gaps between TRBV genes. Using this same set of human genes and an annotated dog reference genome (also curated on IMGT), we performed a similar analysis in IGHV and TRBV gene families between humans and dogs and found similar results.

Given the larger diversity among TRBV genes between these 2 species, we then looked at amino acid diversity in IGHV and TRBV gene families within each of 13 vertebrate species (curated at vgenerepertoire.org), including 5 primates, 6 non-primate mammals, 1 reptile, and 1 fish (Fig 6, Supplementary Text); we again found a similar pattern for each species. The amino acid diversity for each species was calculated between the IGHV genes and between the TRBV genes in that species' reference genome. For all these species, we found that the IGHV gene segments have substantially lower

...

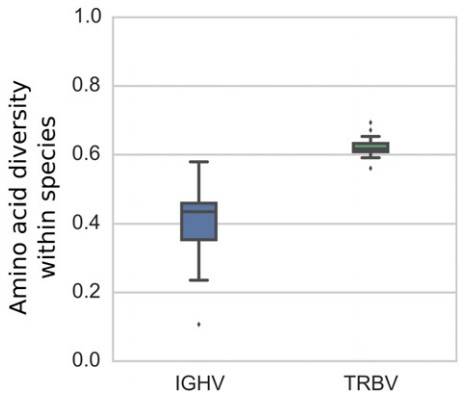

**Figure 6. Box plots of within-species average pairwise diversity of IGHV segments and TRBV segments.**
The results are based on 13 vertebrate species, including 5 primates, 6 non-primate mammals, 1 reptile, and 1 fish (see Supplementary Text for a complete list of species). For each species, pairwise alignments of all pairs of IGHV segments and all pairs of TRBV segments were performed using ssw (Zhao et al, 2013), an implementation of the Smith–Waterman algorithm (Smith & Waterman, 1981).

within-species diversity (about 44%) than the reference TRBV gene segments (about 60% within-species diversity; Fig 6). We also observed less homology between species for the IGHV gene family compared with the TRBV family (Fig S11), which together with the aforementioned lower diversity in IGHV, suggests that IGHV homologs that are shared between species are deleted more frequently than TRBV homologs. This is consistent with our finding that gene duplication and deletion occur more frequently in the IGHV locus. It is possible, however, that rather than being erased, some genes accumulate sufficient amounts of nucleotide changes that cause them to appear as an entirely new gene.

## Discussion

The analysis of gene families remains a technically challenging task in modern genetics. Here, we have made major inroads in quantifying sample-level variation in gene segments in the IGHV and TRBV gene families. We have uncovered patterns of variation that hint at the evolution of the two gene families as well as allelic variants that may be associated with disease specific to a geographic region. Our analysis suggests that the IGHV gene family has experienced more frequent gene duplication/deletion relative to the TRBV gene family over macroevolutionary time scales. The lack of geographical associations for the majority of common copy number polymorphisms in our sample suggests that IGHV and TRBV copy number variation was established early in the history of *Homo sapiens* and that it is unlikely that the presence of particular IGHV or TRBV gene segments is vital against any region-specific pathogens. However, we found a number of alleles in both gene families to be private to a particular region and at nontrivial frequencies. Such allelic variants may be promising candidates for investigating genetic variants that are beneficial against infectious diseases endemic in a geographic region. These differences in IGHV and TRBV may be associated with the different

functions of B cells and T cells, particularly the latter's interaction with major histocompatibility complex molecules, which itself is complex and highly variable.

Our analysis of copy number variation has practical implications for germline IGHV haplotyping: approaches for cataloging variation by sequencing the 1-Mb locus in full (Watson et al, 2013; Steele & Lloyd, 2015) will need to consider the possibility that even in a sample of hundreds of individuals, there will be copy number differences between a substantial fraction of haplotypes. Indeed, we find that when we draw two individuals at random from our sample, there is a 98% chance that they will have different sets of IGHV segments present or absent, but only an 11% chance they will have different sets of TRBV segments present or absent. This calculation is based on a coarser, more robust measure of copy number haplotypes, where we identify each individual by the presence or absence of a segment, and is, therefore, conservative. These numbers remain approximately the same even when we restrict our comparisons to individuals within geographical regions, again indicating that the presence or absence of functional segments does not segregate by geographic region (Table S6). These results provide quantitative support for the conjecture made by Li et al (2002) that "no chromosomes contain the same set of $V_H$ gene segments," where $V_H$ refers to IGHV.

Our results are also of immediate relevance to the adaptive immune receptor repertoire sequencing community. The greater complexity in the IGHV locus suggests that using data analysis methods interchangeably between T-cell receptor sequences and B-cell receptor sequences may not be optimal. Most TRBV genes are operationally distinguishable and appear as a single copy per haplotype. Because T-cell receptors do not undergo further somatic hypermutation, it makes sense to construct the so-called "public" T-cell receptor repertoires and analyze individual repertoires in relation to common public repertoires. In contrast, most IGHV genes either vary in copy number or share long subsequences in common with other genes/pseudogenes/orphon genes in the IGHV family (Supplementary Information). Furthermore, immunoglobulins undergo genetic modification via somatic mutation. The analysis of the antibody repertoire may, therefore, need to be customized to each individual, as suggested by others (Corcoran et al, 2016).

Many challenges remain in genotyping complex and variable regions such as IGHV and TRBV. Our approach of using short-read data has a major advantage in being scalable to large sample sizes, allowing population frequencies to be calculated. However, other approaches may be more appropriate if the goal is to genotype a single individual at base-pair resolution, rather than a large set of individuals at coarser resolution. Another challenge is measuring the rate of nucleotide substitution in IGHV genes, which requires distinguishing between mutations on paralogous regions from true allelic variation. We have adopted a conservative approach here, restricting our calculation to 11 IGHV genes which we are confident are two-copy. However, these 11 genes may not be representative of all the regions of IGHV that are not subject to copy number variation. An approach which can identify larger tracts of IGHV that are structurally conserved across hundreds of individuals will give a better estimate of the nucleotide substitution rate.

## Materials and Methods

### Gene nomenclature

The following sets of gene segments were considered operationally indistinguishable (often more than 95% nucleotide similarity) for our bioinformatic analysis: {*IGHV3-23*, *IGHV3-23D*}, {*IGHV3-30*, *IGHV3-30-3*, *IGHV3-30-5*, *IGHV3-33*}, {*IGHV3-53*, *IGHV3-66*}, {*IGHV3-64*, *IGHV3-64D*}, {*IGHV1-69*, *IGHV1-69D*}, {*IGHV2-70*, *IGHV2-70D*}, {*TRBV4-2*, *TRBV4-3*}, {*TRBV6-2*, *TRBV6-3*}, and {*TRBV12-3*, *TRBV12-4*}.

### SGDP dataset

Whole-genome shotgun sequencing reads were collected in a previous study, the Simons Genome Diversity Project (Mallick et al, 2016). Briefly, 300 genomes from 142 subpopulations were sequenced to a median coverage of 42×, with 100-base pair paired-end sequencing on the Illumina HiSeq2000 sequencers. The reads from 286 of these genomes were mapped to the set of functional alleles (IGHV or TRBV), where our definition of functional is according to the IMGT database annotations (Giudicelli et al, 2005). Of the 286 individuals, only those from non–cell line, that is, blood and saliva DNA sources (109 in total), could be used for IGHV analysis. This is because in these cell lines, which are based on immortalized B cells, the IGHV locus is truncated relative to germline configuration because of VDJ recombination. Details of individual samples can be found in Supplementary Data Table 1 of Mallick et al (2016). For the TRBV locus, we used the full set of 286 genomes, unless otherwise stated. Note: we only had access to 286 genomes of the 300 genomes: 300 minus the 14 individuals with labels SS60044XX.

### Data availability

The raw data for 279 genomes are available through the European Bioinformatics Institute (EBI) European Nucleotide Archive under accession number PRJEB9586. For additional 21 genomes (designated by code Y in the seventh column of Supplementary Data Table 1 in Mallick et al (2016), data are deposited at the European Genome-phenome Archive (EGA), which is hosted by the EBI and the Centre for Genomic Regulation (CRG), under accession number EGAS00001001959. The set of filtered mapped reads used in our study can be found at https://github.com/songlab-cal/SGDP_IGHV_TRBV.

### Read mapping/filtering

For the results above, we used reads mapped to a list of functional IGHV and TRBV (from the online IMGT database Giudicelli et al [2005]). The disadvantage of this procedure is that reads from highly similar pseudogenes and orphon genes may get mixed with reads from functional genes (Figs S12 and S13). Thus, for each of the IGHV and TRBV loci, we filter the set of raw reads, aiming to minimize reads that have been erroneously mapped to a functional gene segment. This required taking into account idiosyncrasies of individual segments, especially their similarity to pseudogenes and orphon genes. We refer the reader to the full details of the filtering steps in the Supplementary Text.

### Copy number calls/contig assembly

After read filtering, we have, for each individual, a set of reads binned by operationally distinguishable segment. We next run the assembler Spades (Bankevich et al, 2012) to construct a contig for each segment to obtain the following:

1. kmer coverage for the segment in that individual.
2. A first estimate of the nucleotide sequence of the individual's gene.

For example, for a fixed individual, the script we execute to assemble the contig for *IGHV6-1* is as follows:

spades.py –k 21 –careful –s IGHV6-1.fastq –o contigs/IGHV6-1.

The choice of kmer of size 21 is because it was the longest kmer that ensured successful contig construction for our 100-bp reads at around 40 coverage depth. The kmer coverage is then converted to per-base coverage, scaled to account for the trapezoidal shape of the read coverage profile, and then normalized by the individual's genome-wide coverage to obtain a point estimate for copy number (details of calculation in Supplementary Text).

### Haplotype phasing and allele/SNV calls

The contigs and reads for two-copy segments were analyzed for allelic and SNVs by phasing these segments for each individual. Because the assembly step in the pipeline produces only one contig, we reconstructed the two distinct allelic sequences on each chromosome through additional steps, which are as follows:

1. Mapped the filtered set of reads to the contig constructed via the customized pipeline using bowtie2 *–local –score-min G,20, 30*.
2. The results from Bowtie2 were fed to GATK (McKenna et al, 2010) for variant calling, producing VCF files identifying polymorphic sites, using the HaplotypeCaller with parameters *-ploidy 2 -stand_call_conf 30 -stand_emit_conf 10*.
3. The variants from GATK were then phased using HapCUT2 (Edge et al, 2017). Procedures for handling instances when HapCUT2 failed are explained in Supplementary Text. To be conservative, we kept only the alleles found in at least two different individuals.

## Supplementary Information

## Acknowledgements

We gratefully acknowledge David Reich for useful discussions and for making short-read data from the Simons Genome Diversity Project available to us. This research is supported in part by an NIH grant R01-GM094402 and a

Packard Fellowship for Science and Engineering. YS Song is a Chan Zuckerberg Biohub investigator.

## Author Contributions

S Luo: conceptualization, data curation, software, formal analysis, investigation, methodology, and writing—original draft, review, and editing.
JA Yu: data curation, software, formal analysis, investigation, methodology, and writing—original draft, review, and editing.
H Li: data curation, software, and methodology.
YS Song: conceptualization, supervision, funding acquisition, investigation, methodology, and writing—original draft, review, and editing.

## Conflict of Interest Statement

The authors declare that they have no conflict of interest.

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
