## [Reviewer comments · Life Science Alliance]

Life Science Alliance

Worldwide genetic variation of the IGHV and TRBV immune receptor gene families in humans

Shishi Luo, Jane Yu, Heng Li, and Yun Song
DOI: <https://doi.org/10.26508/lsa.201800221>

Corresponding author(s): Yun Song, University of California, Berkeley

Review Timeline:

Submission Date:	2018-10-22
Editorial Decision:	2018-11-21
Revision Received:	2019-01-20
Editorial Decision:	2019-02-04
Revision Received:	2019-02-14
Accepted:	2019-02-14

Scientific Editor: Andrea Leibfried

Transaction Report:

November 21, 2018

Re: Life Science Alliance manuscript #LSA-2018-00221-T

Prof. Yun S. Song
University of California, Berkeley
Department of Statistics
321 Evans Hall # 3860
Berkeley, CA 94720-3860

Dear Dr. Song,

Thank you for submitting your manuscript entitled "Worldwide genetic variation of the IGHV and TRBV immune receptor gene families in humans" to Life Science Alliance. The manuscript was assessed by expert reviewers, whose comments are appended to this letter.

As you will see, the reviewers think that your analysis is a valuable contribution to the field, and they provide constructive input on how to further strengthen your work prior to publication. We would thus like to invite you to submit a revised version of your manuscript, addressing the reviewers' individual points. Importantly, please make sure that the statistical power is sufficiently high for drawing conclusions. Furthermore, some clarifications are needed and more details need to be added.

Thank you for this interesting contribution to Life Science Alliance. We are looking forward to receiving your revised manuscript.

Sincerely,

- A letter addressing the reviewers' comments point by point.
- An editable version of the final text (.DOC or .DOCX) is needed for copyediting (no PDFs).
- High-resolution figure, supplementary figure and video files uploaded as individual files: See our detailed guidelines for preparing your production-ready images, <http://life-science-alliance.org/authorguide>
- Summary blurb (enter in submission system): A short text summarizing in a single sentence the study (max. 200 characters including spaces). This text is used in conjunction with the titles of papers, hence should be informative and complementary to the title and running title. It should describe the context and significance of the findings for a general readership; it should be written in the present tense and refer to the work in the third person. Author names should not be mentioned.

B. MANUSCRIPT ORGANIZATION AND FORMATTING:

Full guidelines are available on our Instructions for Authors page, <http://life-science-alliance.org/authorguide>

Reviewer #1 (Comments to the Authors (Required)):

Luo et al. examine the variation of immune "V" gene segments and, in particular, contrast the patterns at the TRBV (T-cell) locus with those at the IGHV (antibody/B-cell) locus. While previous authors have looked at V gene variation, none have done so comprehensively, in part due to

challenges surmounted by the computational pipeline developed by these authors (and previously published). This paper is well written and the conclusions are clear: 1) many alleles at appreciable frequencies in humans are not in the standard IMGT database, 2) the IGHV locus has distinctly higher gene content variation than the TRBV locus, 3) despite the higher content variation, IGHV has slightly lower nucleotide diversity than TRBV. They further look for geographic patterns in gene content as well as allelic variation and find no geographic associations beyond the effect expected from human demography.

While most of the paper is focused on humans, the authors found similar large-scale patterns of TRBV vs IGHV within dogs and a broadly consistent pattern in a limited analysis extending to between species comparisons of 13 vertebrates.

The authors might consider noting in the discussion that the difference between IGHV and TRBV might be due to the latter's interaction with MHC, which is itself a highly variable locus.

Minor points

- pg 5: subheading "IGHV and TRBV have comparable levels of nucleotide diversity" while true set me into thinking they were identical while, as is referenced several times later, TRBV is slightly higher. Better to say this in the subheading as well.
- pg 9: vgenerepertoire.org is misspelled here and also in supplement (currently reads vgenerepOrtoire.org)
- pg 10: typo -- "conjecture made BY Li et al."

Reviewer #2 (Comments to the Authors (Required)):

Comments to the Authors:

Luo et al. explore the genetic variation (copy number and single nucleotide variants) within the human IGHV and TRBV loci using short-read WGS from ~100 subjects for the IGH and almost 300 subjects for the TRBV. Subjects are samples from diverse populations covering worldwide geographic areas.

WGS reads covering the IGHV/TRBV loci were captured by taking the intercept of reads that mapped to the loci in GRCh37 (via bwa) and to IGHV/TRBV gene segments sequences obtained from IMGT (minimap). A filtering procedure is then undertaken to remove reads that are likely derived from pseudogenes and orphon loci (V-like genes from other loci), before using the reads binned by those that map to a given gene segment (or an indistinguishable set of gene segments) to build contigs (spades) that represent the gene segment alleles in each subject. SNVs and haplotypes are then explored as is copy number variation using segment read coverage relative to genome-wide read coverage. Comparative analysis of IGHV and TRBV is undertaken and geographical contributions to genetic diversity considered.

The methodologies described in the manuscript highlight an interesting utilisation of WGS for the exploration of features of the TRBV and IGHV loci in cohorts numbering hundreds of subjects.

Major comments

The analysis concludes that there were no geographical specific differences with respect to copy number. For the IGHV, the analysis was restricted to 109 subjects across the 7 regions. Referring to

Figure S5, was there suitable statistical power for these comparisons given the number of subjects that were studied from each region? Furthermore, the interaction between genotype calling and the genes that are grouped into the indistinguishable sets isn't immediately clear. For example, Figure S5A explores three genotypes that include IGHV3-64, which is grouped with IGHV3-64D for analysis, yet two of the three genotypes include both these genes. How were these distinguished at the genotyping level? Furthermore, why is it expected that VDJ recombination doesn't alter IGHV1-69 polymorphisms? Is it because of the gene location in loci?

It is not clear what is seeking to be correlated with respect to 'copy number polymorphisms'. Is it whether carrying one gene segment in CNV is likely to result in other segments to also be present in the haplotype as CNVs?

IGHV and TRBV are found to have undergone different rates of diversification with regards to gene duplication/deletion, however, for gene segments that are generally heterozygous in most subjects (if that is the correct interpretation of 'two-copy' segments) the SNV diversity is approximately equal for TRBV and IGHV. Is it the read mapping ambiguity that leads to the lower number of IGHV used in this analysis (11) compared to the number of TRBV segments (40)? How is matching to IMGT alleles done (for example, IgBLAST)? Does the increased exclusion of IGHVs impact at all on the potential result here?

Novel alleles are reported from the contigs built from the captured reads. This reinforces the incompleteness of the reference datasets within the field. Coverage details for the polymorphic positions would be helpful in interpreting confidence in these novel alleles. Or at least publication of the read data that supports the inferences. Interestingly, some SNVs were found to be exclusive to particular geographical regions (it would be of interest to be more explicit in stating the number of subjects from each region that were analysed and the number of those that have a given SNV). Overall the manuscript presents a comprehensive approach to the study of polymorphic and polygenic gene families from short read whole genome sequence data from large cohorts.

Minor comments

Introduction

A reference or references in support of the opening sentence of the Introduction regarding CNV would be useful.

The description of IGHV family and TCRB family in the first paragraph of the introduction may cause some confusion with respect to the IG and TR nomenclature where gene families are the groupings of segments that share > 75% sequence identity; for example, the IGHV1, IGHV2, etc families. First sentence of paragraph two of the Introduction "short-read sequencing data" may be better stated as "short-read whole genome sequencing data" to clearly differentiate for the reader that this study is of WGS rather than the usual targeted amplification of gDNA or cDNA that is more common within the field.

Is the reference for the first sentence of paragraph two accurate? This is the IMGT reference dataset, is the reference to cite that the genes are based on IMGT versus the pipeline being based on the reference? If it is the latter, some rewording may improve clarity.

Please expand on how an understanding of the relative abundance of germline IGHV/TRBV gene segment variants can assist in the determination of clonal lineages from VDJ rearrangements as described in the final paragraph of the Introduction. This is a within subject inference that is usually focussed on the CDR3 sequences of which the V segments contribute only a few residues, as such it would be useful to explain on how population level frequencies would improve these inferences.

Results

The nomenclature definitions at the start of the Results are useful. Perhaps your definition of 'functional' should also be included here. The analysis is stated to focus on functional gene segments. No definition of functionality is provided beyond a citation for the IMGT database. Is it therefore the case that it is according to the IMGT annotation of the gene segments? This should

perhaps be stated more explicitly, and consideration made of using this definition over others such as the presence of suitable RSS surrounding the gene segment from genomic contigs, or the observation of the gene segment within VDJ rearrangements? For inferred alleles, how is functionality ascribed? Also is it possible that the variations that you are exploring alter the functionality in a way that may rescue pseudogenes and non-functional genes that you have specifically excluded?

Supplemental Information Text

What versions of the IMGT reference datasets were used for the analysis? Was any curation undertaken to remove truncated sequences or those that include degenerate nucleotides. For example, IGHV1-69*07, is flagged as function but is partial in both the 5' and 3'. Or, IGHV1-2*03, which is flagged as functional but includes an 'n' nucleotide within its sequence.

Further to this, were all allelic variants used, or only a single representative allele for each gene segment? Perhaps, it would be useful to provide the set of IGHV/TRBV that you used for minimap step.

What version of bwa was used for the analysis?

minimap has been deprecated since late September 2017, are there likely to be any challenges in reproducing the analysis using the new minimap2?

I am a little confused by the comments in the final paragraph of the 'Read Mapping' subsection. My understanding is that if you use bwa to map to the reference genome then you fail to capture reads that aren't in the reference (because you can't capture what isn't included), so you add further gene segments using the IMGT functional gene list and capture further reads for these genes using minimap. The part I don't understand is the comment regarding differences in mapping algorithms between different sets - is this referring to difference in bwa and minimap, or to difference between WGS reported in this study and your previous study? Equally, for the caption for Figure S13, this makes it sound as though it is surprising that gene segments missing from the reference are biased against. Was the expectation that these genes would be similar enough to reference genes that the reads would still be captured and data on the gene segment obtained even though it isn't in the reference? If so, this could be stated more explicitly.

Might be of interest to include the full list of the genes that are in this set that are absent from GRCh37 (rather than just the two examples listed in the final sentence of final paragraph of 'Read Mapping').

Are there potential consequences to only grouping genes with the same IGHV family by IMGT nomenclature into the operationally indistinguishable groups? For genes that are highly similar but which have been assigned to different IGHV families, what is the consequence of the multi-mapping for 100bp reads across these?

Is it possible to resolve some of the indistinguishable genes by include the RSS elements (or other upstream/downstream genomic sequence) in the reference sets that are used to capture reads? This would still lead to higher instances of multi-mapping for the parts of the gene segment that are shared, but it may permit a call on which of the 'indistinguishable' segments are detected within a single subject.

General

The Journal data requirements state that the data should be freely available in an appropriate public database. While this may be impractical for the complete WGS from the large cohort, perhaps the subset of reads that were mapped from the WGS to either GRC37 or the curated set of IGHV/TRBV could be uploaded to a suitable database?

Reviewer #3 (Comments to the Authors (Required)):

In their present manuscript, Luo et al. use data sets from Simons Genome Diversity Project to infer

IGHV and TRBV variants, CNVs and haplotypes using an interesting and innovative approach to quantify and correct read mapping. With this the authors address an important and technically challenging aspect in the field of immunogenetics.

While the authors do consider several important technical aspects (e.g. excluding data sets derived from EBV-transformed lymphoblasts from IGHV inference), there is a substantial amount of filtering/grouping steps performed and described in the text. While most of these steps are likely justified in isolation, their combination makes it difficult to comprehend their combined impact on the results. Most prominently among those is the filtering for segments/alleles present in the IMGT database. The manuscript would profit from a step-by-step validation of the processing pipeline, as it appears to be the central point of the manuscript and should be strengthened.

In contrast, the rather far-reaching conclusions made in the Discussion are not sufficiently supported by the data.

Finally, putative novel alleles should be submitted to Genbank/ENA and the Inferred Allele Review Committee (IARC, <https://ogrdp.airr-community.org>).

Response to Reviews

“Worldwide genetic variation of the IGHV and TRBV immune receptor gene families in humans” by Shishi Luo, Jane A. Yu, Heng Li, and Yun S. Song

Manuscript #: LSA-2018-00221-T

We thank the reviewers for their comments, which helped to improve our manuscript. Please find below our point-by-point response. (Our response is shown in blue)

Reviewer #1 (Comments to the Authors (Required)):

Luo et al. examine the variation of immune "V" gene segments and, in particular, contrast the patterns at the TRBV (T-cell) locus with those at the IGHV (antibody/B-cell) locus. While previous authors have looked at V gene variation, none have done so comprehensively, in part due to challenges surmounted by the computational pipeline developed by these authors (and previously published). This paper is well written and the conclusions are clear: 1) many alleles at appreciable frequencies in humans are not in the standard IMGT database, 2) the IGHV locus has distinctly higher gene content variation than the TRBV locus, 3) despite the higher content variation, IGHV has slightly lower nucleotide diversity than TRBV. They further look for geographic patterns in gene content as well as allelic variation and find no geographic associations beyond the effect expected from human demography.

While most of the paper is focused on humans, the authors found similar large-scale patterns of TRBV vs IGHV within dogs and a broadly consistent pattern in a limited analysis extending to between species comparisons of 13 vertebrates.

The authors might consider noting in the discussion that the difference between IGHV and TRBV might be due to the latter's interaction with MHC, which is itself a highly variable locus.

Minor points

- pg 5: subheading "IGHV and TRBV have comparable levels of nucleotide diversity" while true set me into thinking they were identical while, as is referenced several times later, TRBV is slightly higher. Better to say this in the subheading as well.
- pg 9: vgenerepertoire.org is misspelled here and also in supplement (currently reads vgenerepOrtoire.org)

- pg 10: typo -- "conjecture made BY Li et al."

We thank the reviewer for their encouraging feedback and helpful suggestions. We have fixed the two typos and incorporated the reviewer's suggestion to mention interaction with MHC molecules:

“Our analysis suggests that the IGHV gene family has experienced more frequent gene duplication/deletion relative to the TRBV gene family over macro-evolutionary time scales. [...] These differences in IGHV and TRBV may be associated with the different functions of B cells and T cells, particularly, the latter's interaction with major histocompatibility complex molecules, which itself is complex and highly variable.”

We appreciate the reviewer's suggestion to change the subheading “IGHV and TRBV have comparable levels of nucleotide diversity”; however, the differences between measurements of nucleotide diversity are not statistically significant between IGHV and TRBV. As such, we are wary of making claims that are not sufficiently supported by our data, which suggests that the levels of diversity are at least comparable.

Reviewer #2 (Comments to the Authors (Required)):

Comments to the Authors:

Luo et al. explore the genetic variation (copy number and single nucleotide variants) within the human IGHV and TRBV loci using short-read WGS from ~100 subjects for the IGH and almost 300 subjects for the TRBV. Subjects are samples from diverse populations covering worldwide geographic areas.

WGS reads covering the IGHV/TRBV loci were captured by taking the intercept of reads that mapped to the loci in GRCh37 (via bwa) and to IGHV/TRBV gene segments sequences obtained from IMGT (minimap). A filtering procedure is then undertaken to remove reads that are likely derived from pseudogenes and orphon loci (V-like genes from other loci), before using the reads binned by those that map to a given gene segment (or an indistinguishable set of gene segments) to build contigs (spades) that represent the gene segment alleles in each subject. SNVs and haplotypes are then explored as is copy number variation using segment read coverage relative to genome-wide read coverage. Comparative analysis of IGHV and TRBV is undertaken and geographical contributions to genetic diversity considered.

The methodologies described in the manuscript highlight an interesting utilisation of WGS for the exploration of features of the TRBV and IGHV loci in cohorts numbering hundreds of subjects.

We are encouraged that the reviewer finds the work interesting, and we address the reviewer's concerns below, point-by-point. We believe the resulting changes have significantly improved the clarity of the manuscript.

First, we would just like to clarify that our results are based on whole-genome sequencing reads that were mapped via minimap to IGHV/TRBV gene segments defined in IMGT. Initially, we had looked at reads that were mapped via bwa to the IGHV and TRBV loci in GRCh37. However, as explained in detail in the Supplementary Information, the reads mapped to GRCh37 exhibited reference bias.

We have revised the relevant section of the Supplementary Information to make this clear.

Major comments

The analysis concludes that there were no geographical specific differences with respect to copy number. For the IGHV, the analysis was restricted to 109 subjects across the 7 regions. Referring to Figure S5, was there suitable statistical power for these comparisons given the number of subjects that were studied from each region?

The reviewer raises an important point, and indeed some of these regions do not have large enough sample sizes for conclusions to be made. We have added text to caution interpretation of the results from the regions with small sample sizes:

“We note, however, that the sample sizes for the IGHV analysis East Asians, Oceanians, and Native Americans do not have suitable statistical power, and are included for comprehensiveness and illustrative purposes.”

Furthermore, the interaction between genotype calling and the genes that are grouped into the indistinguishable sets isn't immediately clear. For example, Figure S5A explores three genotypes that include IGHV3-64, which is grouped with IGHV3-64D for analysis, yet two of the three genotypes include both these genes. How were these distinguished at the genotyping level?

We regret that our figures may have been misleading, and we thank the reviewer for alerting us to this issue. We indeed cannot technically distinguish these two genes. The arrangement of IGHV3-64 and IGHV3-64D in Figures 2 and 3 as well as Figures S5 and S6 were based on prior work, and not the short read data. For example, from the caption of Figure 2:

“Schematics in the left column show the genomic configuration of these polymorphisms determined by previous studies (Boyd et al., 2010; Chinge et al., 2005; Cho, Wang, Zhao, Carson, & Chen, 1997; Milner, Hufnagle, Glas, Suzuki, & Alexander, 1995; Pramanik & Li, 2002; Sasso, Dijk, Bull, & Milner, 1993; Sasso, Johnson, & Kipps, 1996; Watson et al., 2013)”.

The only information from the short read data that informs these polymorphisms (the left hand panels of Figure 2 and 3) is the copy number. The phasing and distinction of indistinguishable genes is our best estimate of the genomic configuration based on existing literature.

We have modified the captions to be clearer on this point:

Figure 2: “Schematics in the left column show the polymorphisms while the right column displays the relative abundance (RA) in the sample of 109 individuals. Our data inform the copy number of these genes, while the genomic configuration is our best estimate based on previous studies (Boyd et al., 2010; Chinge et al., 2005; Cho, Wang, Zhao, Carson, & Chen, 1997; Milner, Hufnagle, Glas, Suzuki, & Alexander, 1995; Pramanik & Li, 2002; Sasso, Dijk, Bull, & Milner, 1993; Sasso, Johnson, & Kipps, 1996; Watson et al., 2013).”

We have done similarly for Figure 3, Figure S5, and Figure S6.

Furthermore, why is it expected that VDJ recombination doesn't alter IGHV1-69 polymorphisms? Is it because of the gene location in loci?

That is correct. As mentioned in the caption of Figure 2:

“For the polymorphism involving IGHV1-69, we also show the relative abundances in the full sample of 286 individuals in parentheses. This is because the impact of VDJ recombination on cell line samples in this J-distal part of the locus has a negligible effect on our copy number estimates.”

We have rephrased to make this clearer. The second sentence has been replaced by:

“This is because IGHV1-69 and IGHV2-70 are located in the J-distal part of the IGHV locus, making them less likely to be affected by VDJ recombination. Unlike the IGHV polymorphisms that are closer to the J region, we saw negligible differences in copy number estimates for these gene segments in the saliva versus cell-line samples.”

It is not clear what is seeking to be correlated with respect to 'copy number polymorphisms'. Is it whether carrying one gene segment in CNV is likely to result in other segments to also be present in the haplotype as CNVs?

Yes, or more broadly speaking, it illustrates the correlation between the copy number of one gene to that of another gene or genes. More specifically, Figure S15 would indicate that 0 copies of IGHV1-8 correlates strongly with 0 copies of IGHV3-9 and two copies of IGHV5-10-1.

IGHV and TRBV are found to have undergone different rates of diversification with regards to gene duplication/deletion, however, for gene segments that are generally heterozygous in most subjects (if that is the correct interpretation of 'two-copy' segments) the SNV diversity is approximately equal for TRBV and IGHV. Is it the read mapping ambiguity that leads to the lower number of IGHV used in this analysis (11) compared to the number of TRBV segments (40)?

Two-copy genes do not need to be heterozygous. They are actually any genes that are “two copies in the vast majority of individuals in our sample and for which there is minimal read-mapping ambiguity”. By genes with minimal read-mapping ambiguity, we mean genes that do not share subsequences with other known genes (functional or non-functional). There were 10 more IGHV genes than TRBV genes that were predominantly two copies but were excluded from the analysis because they shared subsequences with other genes. In that sense, potential “read mapping ambiguity” reduced the number of analyzed genes that are predominantly two

copies from 24 to 11. To include these additional 13 genes which share subsequences with other known IGHV genes, however, would likely lead to noisy inferences of nucleotide.

Overall, though, we observe a copy number distribution with a much larger spread for IGHV than for TRBV, which has a majority of genes that are two-copies. Given that IGHV has undergone many more gene duplication/deletions than TRBV, it is not unreasonable that our findings suggest fewer two-copy functional IGHV genes than two-copy functional TRBV genes.

How is matching to IMGT alleles done (for example, IgBLAST)? Does the increased exclusion of IGHVs impact at all on the potential result here?

We matched all the contigs using IgBLAST with the following command:

```
./igblastn -germline_db_V support_files/igblast_db/imgt_Vgerm -germline_db_J support_files/igblast_db/imgt_Jgerm -germline_db_D support_files/igblast_db/imgt_Dgerm -organism human -query all_contigs.fasta -auxiliary_data optional_file/human_gl.aux -show_translation
```

This is done with the complete IMGT database pertaining to humans, and there is no increased exclusion of IGHV genes than TRBV genes here. To be clear, **the only analysis that is limited to 11 IGHV genes (as opposed to 40 TRBV genes) is the allelic/SNP variation analysis.** This is a post-processing restriction to a specific analysis that has nothing to do with the pipeline or the copy number variation conclusions. We have added this clarification into the manuscript.

Novel alleles are reported from the contigs built from the captured reads. This reinforces the incompleteness of the reference datasets within the field. Coverage details for the polymorphic positions would be helpful in interpreting confidence in these novel alleles. Or at least publication of the read data that supports the inferences.

Please see our related comments at the very end.

Interestingly, some SNVs were found to be exclusive to particular geographical regions (it would be of interest to be more explicit in stating the number of subjects from each region that were analysed and the number of those that have a given SNV).

In regards to the number of subjects from each region, this was specified in the caption of the table: “SNVs and putative novel alleles in our sample of 109 individuals (218 haplotypes) that are private to a geographic region (14 Africans, 31 West Eurasians, 23 Central Asians – Siberians, 2 East Asians, 27 South Asians, 4 Oceanians, 8 Native Americans).”

Overall the manuscript presents a comprehensive approach to the study of polymorphic and polygenic gene families from short read whole genome sequence data from large cohorts.

Minor comments

Introduction

A reference or references in support of the opening sentence of the Introduction regarding CNV would be useful.

We have added a references from Tuzun et al., 2005 and Sudmant et al., 2015.

The description of IGHV family and TCRB family in the first paragraph of the introduction may cause some confusion with respect to the IG and TR nomenclature where gene families are the groupings of segments that share > 75% sequence identity; for example, the IGHV1, IGHV2, etc families.

We thank the reviewer for pointing out this potential point of confusion. Unfortunately, for this section to make sense it is necessary to use the phrase gene family to refer to the sets of IGHV and TRBV genes. We believe the confusion is minimized because we define what we mean by gene family in its first use:

“...copy number variation remains challenging to quantify and analyze. Nowhere is this more true than in genomic regions that contain **gene families: collections of genes formed through the process of duplication/deletion and diversification of contiguous stretches of DNA (Nei & Rooney, 2005)**. Two gene families that are of particular biomedical relevance but for which variation is not well characterized are the immunoglobulin heavy variable (IGHV) family, a 1 Mb locus located on chromosome 14 (Matsuda et al., 1998; Watson et al., 2013), and the T cell receptor beta variable (TRBV) family, a 500 kb locus located on chromosome 7 (Rowen, Koop, & Hood, 1996).”

First sentence of paragraph two of the Introduction "short-read sequencing data" may be better stated as "short-read whole genome sequencing data" to clearly differentiate for the reader that this study is of WGS rather than the usual targeted amplification of gDNA or cDNA that is more common within the field.

We have taken the reviewer's suggestion and changed this to short-read whole-genome sequencing data.

Is the reference for the first sentence of paragraph two accurate? This is the IMGT reference dataset, is the reference to cite that the genes are based on IMGT versus the pipeline being based on the reference? If it is the latter, some rewording may improve clarity.

We thank the reviewer for pointing out this confusion. We have clarified the text to indicate that the pipeline is from previous work (Luo et al., 2016), and the reference alleles are from the IMGT database.

Please expand on how an understanding of the relative abundance of germline IGHV/TRBV gene segment variants can assist in the determination of clonal lineages from VDJ rearrangements as described in the final paragraph of the Introduction. This is a within subject inference that is usually focussed on the CDR3 sequences of which the V segments contribute only a few residues, as such it would be useful to explain on how population level frequencies would improve these inferences.

We appreciate the reviewer's suggestion and while the V genes do only contribute a few base pairs, the D and J genes themselves are not very long. In total, the CDR3 portion of the V gene may constitute a significant percentage of the CDR3, and there are often cases of lineages in which the D and J genes in the CDR3 region are conserved while it is the V gene which distinguishes the clonal lineage. We have added the following text:

“For example, previous work demonstrates that the V genes may contribute a significant proportion of the CDR3, and often times lineages with conserved D and J genes must be distinguished using V gene information (Li et al., 2004). Past methods for CDR3 determination have included integrating over all possible V genes when information was lacking, and taking population-wide frequencies into account would likely improve the accuracy of such methods (Murugan et al. 2012).”

Results

The nomenclature definitions at the start of the Results are useful. Perhaps your definition of 'functional' should also be included here. The analysis is stated to focus on functional gene segments. No definition of functionality is provided beyond a citation for the IMGT database. Is it therefore the case that it is according to the IMGT annotation of the gene segments? This should perhaps be stated more explicitly, and consideration made of using this definition over others such as the presence of suitable RSS surrounding the gene segment from genomic contigs, or the observation of the gene segment within VDJ rearrangements? For inferred alleles, how is functionality ascribed? Also is it possible that the variations that you are exploring alter the functionality in a way that may rescue pseudogenes and non-functional genes that you have specifically excluded?

The reviewer is correct in the understanding that our definition of functionality is according to the IMGT annotation of the gene segments, and we have modified the text to make this explicit:

“The reads from 286 of these genomes were mapped to the set of functional alleles (IGHV or TRBV), where our definition of functional is according to the IMGT database annotations (*Giudicelli et al., 2005*)”

From the IMGT definition of function: “A germline entity (V-GENE, D-GENE or J-GENE) or a C-GENE is functional if the coding region has an open reading frame without stop codon, and if there is no described defect in the splicing sites, recombination signals and/or regulatory elements.”

Based on this definition, there is baked-in consideration for the presence/absence of suitable RSS surrounding the gene segments. As for “observation of the gene segment within VDJ rearrangements,” to include this in our definition of functional would further limit us to rely on the limited previous work involving the analysis of VDJ rearranged germlines, which is likely to be biased to certain genes (*Chimge et al., 2005*). The IMGT database’s definition of function is an intuitive one and allows the pipeline the freedom to identify the more rarely observed alleles.

Additionally, for the alleles that are inferred from our pipeline, we make no determination of functionality--only annotations of truncations (T) or the presence of stop codons (P). This is likely to have alleles that do not appear in VDJ rearrangements, but the reader is free to filter and restrict this list based on previous and future work.

If our pipeline should rescue any pseudogenes, the query against IMGT using IgBLAST should flag for this.

Supplemental Information Text

What versions of the IMGT reference datasets were used for the analysis? Was any curation undertaken to remove truncated sequences or those that include degenerate nucleotides. For example, IGHV1-69*07, is flagged as function but is partial in both the 5' and 3'. Or, IGHV1-2*03, which is flagged as functional but includes an 'n' nucleotide within its sequence. Further to this, were all allelic variants used, or only a single representative allele for each gene segment? Perhaps, it would be useful to provide the set of IGHV/TRBV that you used for minimap step.

The version of IMGT used was version 3.1.3 for IGHV and version 3.1.4 for TRBV. We do not expect the results to change for the newer versions, given that the update descriptions pertain mostly to bug fixes and the addition of new features.

What version of bwa was used for the analysis?

minimap has been deprecated since late September 2017, are there likely to be any challenges in reproducing the analysis using the new minimap2?

The version of bwa used was bwa-0.7.10-r1017-dirty, which is between v0.7.10 and v0.7.11. Regarding minimap2, the significant updates involve incorporation of the option for base-level alignment and support for spliced alignment, as well as computational speed-ups. As such, one could likely faithfully reproduce the analysis considering that base-level alignment is an option capable of being disabled and that spliced alignment updates should not affect WGS alignment.

I am a little confused by the comments in the final paragraph of the 'Read Mapping' subsection. My understanding is that if you use bwa to map to the reference genome then you fail to capture reads that aren't in the reference (because you can't capture what isn't included), so you add further gene segments using the IMGT functional gene list and capture further reads for these genes using minimap. The part I don't understand is the comment regarding differences in mapping algorithms between different sets - is this referring to difference in bwa and minimap, or to difference between WGS reported in this study and your previous study?

We thank the reviewer for raising this confusion. The two mapping procedures were done and analyzed separately. In the end, our main results are based on procedure (ii) which maps to the set of IMGT alleles. We mention procedure (i) in this section so as to clarify why the reference genome was not used in our read mapping.

Equally, for the caption for Figure S13, this makes it sound as though it is surprising that gene segments missing from the reference are biased against. Was the expectation that these genes would be similar enough to reference genes that the reads would still be captured and data on the gene segment obtained even though it isn't in the reference? If so, this could be stated more explicitly.

Currently the caption of S13 reads:

“Fig. S13. IGHV and TRBV gene segments that are not in GRCh37 assembly are systematically missing from the reads collected via read mapping to GRCh37 (blue, ‘bam_mapped’, procedure (i)) compared to reads mapped to functional IMGT alleles (green, ‘imgt_mapped_filtered’, procedure (ii) with additional filtering).”

We agree with the reviewer that, apriori, one has good reason to believe that gene segments missing from the reference will have biased results under read mapping procedure (i). As such, we are not quite sure we understand how the current caption suggests surprise that gene segments missing from the reference are biased against it.

Might be of interest to include the full list of the genes that are in this set that are absent from GRCh37 (rather than just the two examples listed in the final sentence of final paragraph of 'Read Mapping').

We thank the reviewer for this helpful suggestion and have added the following sentence to the final paragraph:

“Such gene segments for the GRCh37 reference genome are IGHV7-4-1, IGHV4-4, IGHV5-10-1, IGHV4-30-2, IGHV4-30-4, IGHV4-38-2, IGHV1-69-2, IGHV3-NL1, TBRV5-8, TRBV6-2, TRBV7-2, TRBV7-7, TRBV7-9, TRBV10-3, TRBV11-3, TRBV12-3, TRBV12-5, TRBV13, TRBV14, TRBV15, TRBV16, and TRBV18.”

Are there potential consequences to only grouping genes with the same IGHV family by IMGT nomenclature into the operationally indistinguishable groups? For genes that are highly similar but which have been assigned to different IGHV families, what is the consequence of the multi-mapping for 100bp reads across these?

The reviewer brings up an interesting point. However, from our understanding, gene families are primarily defined “on the basis of nucleotide sequence homology” (Cook et al., 1994) and there does exist functional genes which are more similar to a gene in a different family than to a gene within the same gene family. This can be seen by the phylogenetic tree reconstruction of the functional IGHV genes in Figure 1 of Luo et al., 2016, which displays substantial separation between the different gene families. Additionally, Figure 2 of Matsuda et al., 1998 shows a phylogenetic tree construction for all the IGHV gene segments. With the exception of pseudogene IGHV4-44.1P, gene segments have significantly more nucleotide similarity to genes belonging to the same family than to genes belonging to different gene families.

Is it possible to resolve some of the indistinguishable genes by include the RSS elements (or other upstream/downstream genomic sequence) in the reference sets that are used to capture reads? This would still lead to higher instances of multi-mapping for the parts of the gene segment that are shared, but it may permit a call on which of the 'indistinguishable' segments are detected within a single subject.

The reviewer makes an excellent suggestion, but we were cautious of using recombination signal sequences because many of the alleles belonging to the same set of indistinguishable gene actually have the same RSS. For example, alleles TRBV12-3*01 and TRBV12-4*01 actually have the same RSS, which doesn't help distinguish the operationally defined gene {TRBV12-3, TRBV12-4}. Additionally, often times the recombination signal sequences are not available for the complete set of alleles for a particular gene. For example, the RSS of IGHV3-33*01, IGHV3-33*02, and IGHV3-33*06 are distinct from the alleles of IGHV3-30, IGHV3-30-3, and IGHV3-30-5, but the RSS of IGHV3-33*06 differs from the other two alleles. Additionally, the IGHV3-33*03, IGHV3-33*04, and IGHV3-33*05 are not available on the IMGT database, but could possibly be the same as one of the alleles of IGHV3-30, IGHV3-30-3, or IGHV3-30-5.

General

The Journal data requirements state that the data should be freely available in an appropriate public database. While this may be impractical for the complete WGS from the large cohort, perhaps the subset of reads that were mapped from the WGS to either GRC37 or the curated set of IGHV/TRBV could be uploaded to a suitable database?

We thank the reviewer for this suggestion. We would be glad to make publicly available the set of filtered and mapped reads that are used in our analysis.

Reviewer #3 (Comments to the Authors (Required)):

In their present manuscript, Luo et al. use data sets from Simons Genome Diversity Project to infer IGHV and TRBV variants, CNVs and haplotypes using an interesting and innovative approach to quantify and correct read mapping. With this the authors address an important and technically challenging aspect in the field of immunogenetics.

While the authors do consider several important technical aspects (e.g. excluding data sets derived from EBV-transformed lymphoblasts from IGHV inference), there is a substantial amount of filtering/grouping steps performed and described in the text. While most of these steps are likely justified in isolation, their combination makes it difficult to comprehend their combined impact on the results. Most prominently among those is the filtering for segments/alleles present in the IMGT database. The manuscript would profit from a step-by-step validation of the processing pipeline, as it appears to be the central point of the manuscript and should be strengthened.

In contrast, the rather far-reaching conclusions made in the Discussion are not sufficiently supported by the data.

Finally, putative novel alleles should be submitted to Genbank/ENA and the Inferred Allele Review Committee (IARC, <https://ogrdp.airr-community.org>).

We are glad the reviewer appreciates the importance of the problem, and finds our approach interesting and innovative.

The reviewer has two major concerns: one pertaining pipeline validation and the other pertaining conclusions made in the Discussion.

Regarding the first concern that the pipeline should have step-by-step validation results, a great deal of parameter tuning and stepwise validation was done as described in our earlier work (Luo et al., 2016). We directly address the reviewer's concern by looking at each step of the pipeline:

1. Map the reads to the set of functional and full length IMGT alleles using bowtie2.

For this step, we simulated reads from GRCh37 with multiple choices of mapping thresholds using bowtie2. Validation was done by comparing the inferred read position in the genome to that of actual position in the genome. As exemplified in Figure S9, the leftmost plot shows a mapping threshold that is too low, allowing for multiple mismatched reads from pseudogenes and similar functional genes. The rightmost plot shows too few

mapped reads, indicating that the parameters used are much too stringent. The center plot shows strong correlation between the inferred and true position. Similar plots were done for the rest of the genes, and the correct parameters were chosen based on this analysis. Additionally, bowtie2 is a popular software for sequence alignment with a number of independent validation studies (Langmead et al., 2012; Hatem et al., 2013; Thankaswamy-Kosalai et al., 2017).

2. Cluster the set of reads according to the operationally defined gene segments.

Combining extremely similar genes and treating them as a newly defined gene was necessary in order to reduce the mapping ambiguity between alleles of different genes. When using these carefully chosen clustering of genes, the mapping ambiguity is decreased as shown in Figure 1B and Figure 1C of Luo et al., 2016. This has the effect of limiting inference power, but does not subject the results to further major inaccuracies, considering that the sequences are highly similar. For example, if an individual has two copies of IGHV3-64 and one copy of IGHV3-64D, our results will indicate 3 copies of operationally defined gene segment {IGHV3-64, IGHV3-64D}. We do not have enough information to distinguish the two genes, but it does not impact the correctness of the total copies called for the operationally defined gene. Additionally, our use of operationally defined gene segments would not affect our allelic/SNP analysis, because these were excluded from the analysis. To be conservative our allelic/SNP analysis is restricted to predominantly two-copy genes that can be robustly called.

3. Apply gene-specific filtering rules to minimize erroneously mapped reads.

The supplementary section ‘Read Filtering’ has explanations and justifications for this step. At a high-level, the idea is to filter out reads from other genes which share long subsequences with the correct gene. In some cases, this was resolved by taking the top unique hit from the read-mapping results, which is a natural and standard protocol for resolving multiple mappings. As mentioned in the text:

“Note that the per-base read coverage, averaged over all 109 individuals (from blood/saliva samples), matches very closely with what is expected theoretically. Specifically, the coverage decreases linearly towards the edges because reads that only partially cover the segment will have lower mapping scores. [...] The per-base read coverage of around 40 is also consistent with the median genomic coverage of 42 across the full Simons sample (Supplementary Data Table 1 of (Mallick et al., 2016)).”

Sometimes, however, if we only assigned the reads with a unique top hit, then we would lose information on functional genes that share exact subsequences. For these cases, we would distribute the reads with uniform probability to all the top hits. This would not introduce mistakes in the assembly, because the subsequence they share is the same, and in the case where the top hits have the same copy number (like the case with 3-74) then it would not bias our copy number estimate, but correct it. For 3 of the roughly 45 genes, where the copy number was not consistent among the top hits, we had to resort to a more complex, randomized approach, but in this situation this was the best estimate possible given our data.

4. Perform de novo read assembly using SPAdes and run the constructed contigs against the IMG2 database using IgBLAST.

SPAdes is a commonly used software for read assembly and has been extensively benchmarked (Bankevich et al., 2012; Sović et al., 2016; Barrero et al., 2017). Likewise, IgBLAST is a highly used tool for searching against germline gene databases that caters specifically to VDJ-rearranging loci and has independent benchmarking results (Ye et al., 2013). Additionally, we performed benchmarking for our gene segment calls and copy number in our previous paper, Luo et al., 2016. From the ‘Method Performance’ section in the current manuscript:

“To summarize, we simulated reads using all combinations of read length (70, 100, 250bp), reference genomes (GRCh37 and GRCh38), and coverage depth (30x, 40x, 50x), and measured the recall, the fraction of operationally distinguishable gene segments that are correctly called by the pipeline. All except 2 of the 18 combinations demonstrated 100% recall, and the remaining 2 simulations had recall of 97%.”

We have also measured the overall performance for hundreds of simulated genomes for both IGHV and TRBV. If the reviewer has specific suggestions on reasonable benchmarking that should be done in addition to what has currently been done, we welcome these explicit suggestions.

Secondly, the reviewer suggests that the conclusions made in the Discussion are not well supported by the data. Unfortunately, we are unclear as to which particular conclusions are not well supported and in what way they are unfounded. Consequently, we examine the conclusions made in the Discussion and address in what way they are well-supported:

- A. IGHV has experienced more frequent duplications/deletions relative to TRBV.

This conclusion is well-supported by our data, as summarized in Figure 1. Additionally, The fact that we find many more copy number polymorphisms in IGHV than TRBV is also a testament to this claim. In fact, the list of copy number polymorphisms for IGHV is not close to complete, because in order to be conservative, we excluded copy number polymorphisms involving genes which did not have well-behaved read profiles. The TRBV genes were predominantly well-behaved, and we expect the list of copy number polymorphisms for TRBV in Figure 3 to be much closer to complete, whereas the opposite is like true for IGHV in Figure 2. Even though Figure 3 is likely more complete, there are many more CNVs in Figure 2 (IGHV), further indicating greater duplication/deletion dynamics in IGHV than TRBV. Previous work such as that done by Matsuda et al., 1998 found 13 DNA sequences ranging from 4-24 kbp which constitute 67% of the entire locus. This is in comparison to 30-47% of the TRBV loci that constitute homologous regions.

- B. A number of alleles in both gene families are found to be private to a particular region and at non-trivial frequencies.

This is directly supported by Table S5, which demonstrates multiple alleles were found to be specific to certain regions. A multitude of SNPs throughout the entire genome have been found to be prevalent in certain geographic regions and several organizations are interested in using geographic-specific SNPs (e.g., Consortium on Asthma among African-ancestry Populations in the Americas). It is entirely reasonable that the IGHV and TRBV loci should have alleles/SNPs that are prevalent in certain populations as well.

- C. Even in a sample of hundreds of individuals, there will be copy number differences between a substantial fraction of haplotypes.

This is evidenced by the fact that “we find that when we draw two individuals at random from our sample, there is a 98% chance that they will have different sets of IGHV segments present or absent, but only an 11% chance they will have different sets of TRBV segments present or absent.” In addition, Table S6 demonstrates this is consistent, even when analyzing specific geographic regions and is supported by a number of previous works: “The VH region may contain large common insertion/deletion polymorphisms, or alternatively, the intergenic sequences in the VH region are highly diversified among the haplotypes.” (Li et al., 2002). This claim is also consistent with the hypothesis that IGHV has experience more frequent duplication/deletions than TRBV.

We would like to reiterate that this is that largest joint study of the IGHV and TRBV loci, and the majority of work prior to this has drawn conclusions from data with much smaller sample sizes. Should the reviewer have specific concerns that remain, we would be happy to address the issue.

February 4, 2019

RE: Life Science Alliance Manuscript #LSA-2018-00221-TR

Prof. Yun S. Song
University of California, Berkeley
Department of Statistics
321 Evans Hall # 3860
Berkeley, CA 94720-3860

Dear Dr. Song,

Thank you for submitting your revised manuscript entitled "Worldwide genetic variation of the IGHV and TRBV immune receptor gene families in humans". As you will see, the reviewers appreciate the introduced changes and we would thus be happy to publish your paper in Life Science Alliance pending final revisions necessary to meet our formatting guidelines:

- please provide accession numbers for the raw data as requested by the reviewers
- please link your profile in our submission system to your ORCID iD, you should have received an email with instructions on how to do so
- please list 10 authors et al in your reference list
- please add callouts in the main manuscript text to figures S13, S15, S16, S17, S18 and S19
- please note that supplementary figures can only be published as single-page files - please either fix or upload current figures S2, S3, S9 and S10 as 'datasets'.

A. FINAL FILES:

-- High-resolution figure, supplementary figure and video files uploaded as individual files: See our detailed guidelines for preparing your production-ready images, <http://life-science-alliance.org/authorguide>

-- Summary blurb (enter in submission system): A short text summarizing in a single sentence the study (max. 200 characters including spaces). This text is used in conjunction with the titles of papers, hence should be informative and complementary to the title. It should describe the context and significance of the findings for a general readership; it should be written in the present tense

and refer to the work in the third person. Author names should not be mentioned.

B. MANUSCRIPT ORGANIZATION AND FORMATTING:

Full guidelines are available on our Instructions for Authors page, <http://life-science-alliance.org/authorguide>

Sincerely,

Reviewer #1 (Comments to the Authors (Required)):

I am satisfied with this version of the paper.

Reviewer #2 (Comments to the Authors (Required)):

Comments to Authors:

Luo et al. revised manuscripts addresses many of the major and minor comments for the original submission. Some comments remained to be addressed:

Major -

No accession number has been provided for the data deposit.

Minor -

I remained sceptical of the value of population level IGHV frequencies for informing on clonal lineages.

Reviewer #3 (Comments to the Authors (Required)):

The majority of my concerns have been addressed.

Raw NGS data and the novel IGHV / TRBV variants should be deposited in a public database prior to publication.

February 14, 2019

RE: Life Science Alliance Manuscript #LSA-2018-00221-TRR

Prof. Yun S. Song
University of California, Berkeley
Department of Statistics
321 Evans Hall # 3860
Berkeley, CA 94720-3860

Dear Dr. Song,

Thank you for submitting your Research Article entitled "Worldwide genetic variation of the IGHV and TRBV immune receptor gene families in humans". It is a pleasure to let you know that your manuscript is now accepted for publication in Life Science Alliance. Congratulations on this interesting work.

DISTRIBUTION OF MATERIALS:

Again, congratulations on a very nice paper. I hope you found the review process to be constructive and are pleased with how the manuscript was handled editorially. We look forward to future exciting submissions from your lab.

Sincerely,
